

# Clinical value of serum polyunsaturated fatty acids in patients with gastric polyps

Nan Li[1], Qiliu Qian[1], Jun Ouyang[2,3], Mingyue Hu[1], Juan Liu[1], Hailu Wu[1], Ruihua Shi[1] and Shiya Zheng[4]

[1] Zhongda Hospital Affiliated to Southeast University, Department of Gastroenterology, Nanjing, China
[2] Zhenjiang Hospital Affiliated to Nanjing University of Chinese Medicine, Zhenjiang, China
[3] Zhenjiang Traditional Chinese Medicine Spleen and Stomach Disease Clinical Medicine Research Center, Zhenjiang, China
[4] Zhongda Hospital Affiliated to Southeast University, Department of Oncology, Nanjing, China

## ABSTRACT

Polyunsaturated fatty acids (PUFAs) are vital nutrients in human physiology and are implicated in various chronic diseases. However, the relationship between PUFAs and gastric polyps remains unclear. This study employed liquid chromatography-tandem mass spectrometry (LC-MS/MS) to assess PUFA levels in the serum of 350 patients, along with analyzing the ω-6 to ω-3 ratio. The results revealed significant differences in the levels of C16:1, C18:1, C18:2, α-C18:3, γ-C18:3, C20:1, C20:4, C20:5, ω-3-C22:5, ω-6-C22:5, and C22:6, as well as ω-6 to ω-3 ratio between the control and gasteic polyp groups. Moreover, setting the threshold for ω-6: ω-3 at 10 revealed a close correlation between polyp occurrence and this ratio. These findings suggest that PUFAs and the ω-6 to ω-3 ratio hold promise as potential early screening markers for gastric polyps. However, further research is imperative to elucidate the underlying mechanisms and therapeutic potential of PUFAs in managing gastric polyps.

## INTRODUCTION

Gastric polyps encompass a spectrum of diseases diseases originating from the mucosal epithelium of the stomach. These polyps, either are sessile or stalked lesions, have the potential for malignant transformation (*Yacoub et al., 2022*). Previous studies have reported a low incidence rate of neoplastic transformation in gastric polyps, estimated at 0.0582%, with progression typically slow and insidious (*Song et al., 2015*; *Forte et al., 2020*). Despite most gastric polyp cases being asymptomatic, they are typically diagnosed through procedures such as upper endoscopy or barium radiograph. Hence, early detection and diagnosis are necessary for preventing the development of gastric cancer.

The occurrence and development of malignancy in gastric polyps is closely related to dietary intake, including calories intakes, proteins, carbohydrates, and various fatty acids (*Zhang et al., 2020*). Previous studies have indicated that ω-6 polyunsaturated fatty acids (PUFAs) may promote inflammation, enhance tumor angiogenesis, and facilitate

Corresponding author
Shiya Zheng, zhengshiya1990@163.com

tumorigenesis, while ω-3 PUFAs exert protective effects. Moreover, proteins and enzymes involved in fatty acid metabolism have been identified as potential biomarkers of early gastric cancers (*Jiang et al., 2017*). In clinical practice, distinct distribution trends of PUFAs in patients with gastric polyps compared to those without polyps. However, the precise role of PUFAs in gastric polyps remains poorly understood.

Considering these findings, this study, employs liquid chromatography-tandem mass spectrometry to assess serum PUFA levels in patients with gastric polyps. Additionally, we discuss the effects of PUFAs detection in patients with gastric polyps from the perspective of endoscopic and pathological diagnosis.

## METHODS

### Study design and patients

A retrospective study was conducted in the Digestive Endoscopy Center of Zhongda Hospital affiliated to Southeast University, spanning from April 2019 to April 2020. A total of 350 serum samples were obtained from enrolled participants, who had not undergone fatty acids-related treatments previously. Based on endoscopy reports, participants were categorized into two groups: the gastric polyp group (patients with gastric polyps), and the control group (patients without gastric polyps). Within the gastric polyp group, patients were further subdivided into two subgroups based on the number and pathological type of polyps: single polyp subgroup and multiple polyps subgroup, hyperplastic polyps subgroup, and fundic gland polyps subgroup.

All participants signed informed consent, and the study was approved by the Ethics Committee of Zhongda Hospital affiliated to Southeast University (Grant No. 2023ZDSYLL100-P01).

Definition according to ICD-11:

Gastric polyp: A protruding lesion on the gastric epithelium resulting from local overgrowth of gastric epithelial cells, classified as pedunculated or sessile.

Fundic gland polyp of the stomach: Arising from the overgrowth of gastric foveolar epithelial cells. Hyperplastic polyps typically develop against a background of atrophic gastritis, often associated with long-term *Helicobacter pylori* infection.

Hyperplastic polyp of the stomach: Resulting from the overgrowth of gastric foveolar epithelial cells, often associated with atrophic gastritis and long-term *H. pylori* infection.

### PUFAs detection and analysis

The levels of 11 types of fatty acids (C16:1 (palmitoleic acid), C18:1 (oleic acid), C18:2 (linoleic acid), α-C18:3 (alpha-linolenic acid, ALA), γ-C18:3 (γ-linolenic acid), C20:1 (eicosenoic acid), C20:4 (arachidonic acid, AA), C20:5 (eicosapentaenoic acid, EPA), ω-3-C22:5 (docosapentaenoic acid (n-3)), ω-6-C22:5 (docosapentaenoic acid (n-6)), and C22:6 (docosahexaenoic acid, DHA)) were measured using liquid chromatography-tandem mass spectrometry with an API 3200 LC-MS/MS system (AB SCIEX Ltd., Framingham, MA, USA). The blood samples were collected and stored in blood collection tubes without additives or coagulants. The derivatization of fatty acids was performed using a polyunsaturated fatty acid assay kit (ClinMeta Co, Ltd, Shanghai, China) following the

manufacturer's instructions. The ratio of ω-6 to ω-3 was calculated as the ratio of ω-6 group PUFAs (including C18:2, γ-C18:3, AA, and ω-6-C22:5) to ω-3 group PUFAs (including ALA, EPA, ω-3-C22:5, and DHA).

### Statistical analysis

SPSS software (version 26, Chicago, IL, USA) was used to conduct statistical analyses. Measurement data were expressed as the mean ± standard deviation. The t-test was utilized for comparing different groups, while the Chi-square test was used for categorical variables. A $P < 0.05$ was considered statistically significant.

## RESULTS

A total of 350 participants from our center were enrolled in this study, comprising 161 patients in the gastric polyp group and 189 in the control group (Fig. 1). Table 1 presents the basic characteristics of the participants along with the statistical analysis of PUFA expression levels and the ω-6/ω-3 ratio. The levels of 10 PUFAs, except for C20:5 ($P = 0.09$), were significantly different ($P < 0.05$, Table 1), and a similar trend was observed for the ω-6/ω-3 ratio ($P = 0.01$). Previous research has demonstrated that a ratio of ω-6: ω-3 between 4 and 10 was conducive to health. Our findings indicate that when the critical value for the ω-6: ω-3 ratio was set at 10 (Table 2), the occurrence of gastric polyps was strongly associated with this ratio ($P < 0.001$).

Moreover, nearly half of the patients presented with single polyps ($n = 85$, 52.80%). However, there was no significant difference in PUFA levels and the ω-6 to ω-3 ratio between the single polyp subgroup and multiple polyps subgroup ($P = 0.301$). Histopathological diagnosis was performed for 92 patients with polyps, revealing hyperplastic polyps ($n = 134$, 83.23%) and fundic gland polyps ($n = 27$, 16.77%). Notably, no significant difference in PUFA levels was observed between these two subgroups ($P = 0.577$).

## DISCUSSION

PUFAs constitute essential nutritional elements in human metabolism, exerting significant roles in diseases through regulation of immunomodulatory effects, inhibition of inflammation, and attenuation of apoptosis (*Avila et al., 2022*; *Chen et al., 2022*, *2023*). LC-MS/MS is a selective and sensitive method for metabolite concentration detection (*Serafim et al., 2019*). Nevertheless, the application of PUFA detection in gastric polyps remains underreported. Our study reveals specific expression of 10 PUFAs (except for C20:5) in patients with gastric polyps.

ω-6 PUFAs, particularly arachidonate, serve as precursors of pro-inflammatory mediators, inducing low-grade inflammation, oxidative stress, endothelial dysfunction, and atherosclerosis (*DiNicolantonio & O'Keefe, 2018*; *Marklund et al., 2019*). Additionally, ω-6 PUFAs can promote malignant transformation progression. Studies on angiogenesis in gastric cancer have highlighted the role of ω-6 PUFAs in enhancing angiogenesis through the COX2- PGE2 pathway, facilitating vascular endothelial cell proliferation and invasion. Hence, an increase in the levels of ω-6 PUFAs suggests a specific response in
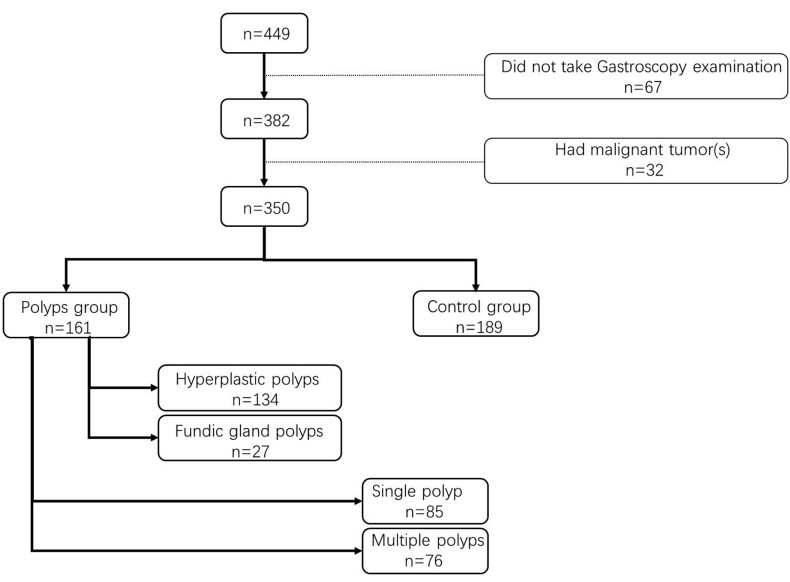

**Figure 1 Flow chart of the research.**

**Table 1 The levels of PUFAs and ω-6/ω-3 in different groups.**

|  | Polyp group | Control group | T test | P |
|---|---|---|---|---|
| Gender | 71M/90F | 100M/89F |  |  |
| Age | 55.24 ± 10.93 | 53.36 ± 12.14 |  |  |
| C16: 1* | 16.04 ± 10.79 | 18.45 ± 11.26 | −2.04 | 0.04 |
| C18: 1 | 229.86 ± 121.82 | 283.75 ± 143.54 | −3.75 | <0.01 |
| C18: 2 | 217.03 ± 132.94 | 276.51 ± 1,311.33 | −4.20 | <0.01 |
| α-C18: 3 | 10.63 ± 6.74 | 14.05 ± 8.91 | −3.99 | <0.01 |
| γ-C18: 3 | 2.07 ± 1.43 | 2.58 ± 1.60 | −3.12 | <0.01 |
| C20: 1 | 2.02 ± 1.51 | 2.85 ± 1.82 | −4.56 | <0.01 |
| C20: 4 | 38.74 ± 42.22 | 53.20 ± 46.75 | −3.01 | <0.01 |
| C20: 5 | 3.00 ± 3.76 | 3.70 ± 3.93 | −1.69 | 0.09 |
| ω-3-C22: 5 | 2.81 ± 2.50 | 3.89 ± 2.59 | −3.94 | <0.01 |
| ω-6-C22: 5 | 1.70 ± 1.77 | 2.40 ± 1.88 | −3.58 | <0.01 |
| C22: 6 | 16.46 ± 15.12 | 22.44 ± 15.54 | −3.63 | <0.01 |
| ω-6/ω-3 | 9.61 ± 4.24 | 8.47 ± 3.56 | 2.70 | 0.01 |

**Note:**[*]
[*] All the concentration units of the PUFAs are μmol/L.

patients with gastric polyps, potentially contributing to its progression (*Marklund et al., 2019*). Our study indicates a statistical difference in ω-6 PUFA levels between patients with gastric polyps and those without polyps.

Moreover, two crucial PUFAs from the ω-3 group, EPA and DHA, increased levels in patients with gastric polyps. These PUFAs, sourced from fish fat, are essential components of cell membranes and play vital roles in anti-inflammatory and anti-hyperlipidemia process (*Marklund et al., 2019*; *Ye & Ghosh, 2018*). The accumulation of PGE3, an intermediate product in ω-3 metabolism, inhibits the proliferation and invasion of vascular

**Table 2 The value of ω-6/ω-3 in the gastric polyps group and the control group.**

| ω-6/ω-3 | Polyp group(n, %) | | Control group (n, %) | | |
| --- | --- | --- | --- | --- | --- |
| <10 | 99 | 38.51% | 148 | 78.31% | P < 0.001 |
| ≥10 | 62 | 61.49% | 41 | 21.69% | |
| <5 | 11 | 6.83% | 13 | 20.11% | P = 1.000 |
| ≥5 | 150 | 93.17% | 176 | 93.12% | |
| <4 | 2 | 1.24% | 1 | 6.88% | P = 1.000 |
| ≥4 | 159 | 98.76% | 188 | 99.47% | |

endothelial cells, thereby inhibiting the angiogenesis of gastric tumors (*Ma et al., 2017*). EPA and DHA also inhibit proliferation and induce apoptosis by disrupting cell metabolism. The downregulation of ω-3 PUFAs in erythrocytes significantly correlates with gastric carcinogenesis (*Sheng et al., 2016*). Furthermore, the serum level of DHA was significantly lower compared to healthy individuals. Accompanied by malignant potential, our study revealed that EPA and DHA levels in patients with gastric polyps were upregulated. Moreover, without exogenous uptake, the high expression of EPA and DHA was associated with the compensatory acceleration of α-C18:3 (ALA) metabolism. Under normal conditions, ALA could be transformed into EPA and DHA; however, the efficiency of this process is low (4–8% in males, 9.2–21% in females) (*Calder, 2015*).

Higher levels of EPA and DHA transformed from ALA exert anti-tumor and anti-inflammation, further reducing the possibility of malignancy. Moreover, owing to this dysregulation, the ω-3 PUFAs decrease significantly in patients with gastric cancer. Accordingly, a study speculated that if DHA intake is increased post-transformation interruption, the risk of gastric cancer could be significantly reduced (*Lee et al., 2018*). This observation suggests a potential role for monitoring serum ω-3 unsaturated fatty acid levels in diagnosing and preventing malignant transformation in gastric polyps. Notably, in this study, the median levels of EPA in patients with gastric polyps were similar to that of the control, while ω-3 unsaturated fatty acids showed statistical differences between the groups.

The metabolism of ω-3 and ω-6 PUFAs involves shared enzymes, leading to competitive inhibition. For instance, EPA and DHA can be metabolized into prostaglandin E3, thromboxane A2, and human leukotriene B5, pathways that AA synthesis and metabolism. (*Fan et al., 2003*). This dynamic balance plays a critical role in disease development and progression. ω-3 PUFAs have been reported to reduce tumor volume and number by downregulating eicosanoids derived from AA *Irun, Lanas & Piazuelo (2019)*.

Hence, the ratio of ω-6 to ω-3 unsaturated fatty acid is widely used in evaluating the metabolic status of unsaturated fatty acids. Animal experiments have demonstrated that a low ω-6/ω-3 ratio confers protective effects on the intestinal mucosa, while a higher ratio correlates with severe inflammation (*Tian et al., 2013*). In our study, the ω-3/ω-6 ratio in patients with gastric polyps was significantly lower than that in healthy individuals, reflecting the upregulation of both ω-6 and ω-3 PUFAs in gastric polyps, which can be

attributed to the stronger protective effect of the ω-3 PUFAs. Thus, during the progression of gastric polyps, the protective effect of PUFAs speculated to be more dominant.

Previous studies have primarily focused on the relationship between cardiovascular disease and PUFAs and dietary fatty acid intake. Data suggests that when the ω-6/ω-3 ratio ranges between 4:1 to 10:1, the risk of various heart diseases including myocardial infarction would be significantly decreases (*Simopoulos, 2002*). Based on this data, our study further investigated the ω-6/ω-3 ratio by dividing it into three groups (<4, <5, and <10). We found that under a higher value (<10), more patients exhibited an ideal ω-6/ω-3 ratio. However, applying the 4:1 and 5:1 ratios did not yield significant differences between the gastric polyps group and the control group, indicating that even with compensatory downregulation of ω-6/ω-3, the ideal ratio may not be fully achieved.

Routine testing of PUFA levels and the ω-6/ω-3 ratio in all gastroenterology department patients, particularly those with a family history of gastric cancer, is recommended to aid in the early detection of gastric polyps.

However, our study has limitations. It is a retrospective and cohort-based study, with a limited number of participants from a single center, potentially leading to selection bias. Future studies should include a larger sample size to further observe changes in the unsaturated fatty acid profile in patients with gastric polyps.

In conclusion, this study measured the levels of PUFAs in patients with gastric polyps and identified characteristics features of PUFAs, especially the ω-6/ω-3 ratio. Future studies should include a larger sample size to explore the unsaturated fatty acid profile changes.

### Funding
This work was supported by the National Natural Science Foundation of China Youth Foundation (81302162) and the Open Project of Zhenjiang Traditional Chinese Medicine Spleen and Stomach Disease Clinical Medicine Research Center (Zhenjiang Hospital Affiliated to Nanjing University of Chinese Medicine) (No. SSPW2022-KF06).
The funders had no role in study design, data collection and analysis, decision to publish, or preparation of the manuscript.

### Grant Disclosures
The following grant information was disclosed by the authors:
National Natural Science Foundation of China Youth Foundation: 81302162.
Open Project of Zhenjiang Traditional Chinese Medicine Spleen and Stomach Disease Clinical Medicine Research Center (Zhenjiang Hospital Affiliated to Nanjing University of Chinese Medicine): SSPW2022-KF06.

### Competing Interests
The authors declare that they have no competing interests.

## Author Contributions

- Nan Li performed the experiments, authored or reviewed drafts of the article, and approved the final draft.
- Qiliu Qian performed the experiments, analyzed the data, authored or reviewed drafts of the article, and approved the final draft.
- Jun Ouyang analyzed the data, prepared figures and/or tables, and approved the final draft.
- Mingyue Hu performed the experiments, prepared figures and/or tables, and approved the final draft.
- Juan Liu analyzed the data, prepared figures and/or tables, and approved the final draft.
- Hailu Wu analyzed the data, prepared figures and/or tables, and approved the final draft.
- Ruihua Shi conceived and designed the experiments, authored or reviewed drafts of the article, and approved the final draft.
- Shiya Zheng conceived and designed the experiments, authored or reviewed drafts of the article, and approved the final draft.

## Human Ethics

The following information was supplied relating to ethical approvals (*i.e.*, approving body and any reference numbers):

The study was approved by the Clinical Research Ethical Committee of Zhongda Hospital Affiliated to Southeast University (Grant No. 2023ZDSYLL100-P01).

## Data Availability

The raw measurements are available in the Supplemental Files.

## Supplemental Information

Supplemental information for this article can be found online at http://dx.doi.org/10.7717/peerj.17413#supplemental-information.

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
