# Peer review of "Clinical value of serum polyunsaturated fatty acids in patients with gastric polyps"

_PeerJ, doi:10.7717/peerj.17413_

## Round 0.1 · original submission · Major Revisions

Please respond to all reviewer's comments and resubmit the revised manuscript with track changes. Also please submit the point-by-point response to the reviewer's comments.

**Language Note:** The review process has identified that the English language must be improved. PeerJ can provide language editing services - please contact us at copyediting@peerj.com for pricing (be sure to provide your manuscript number and title). Alternatively, you should make your own arrangements to improve the language quality and provide details in your response letter. – PeerJ Staff

Reviewer 1 ·

Basic reporting

Li et al reported results from the study titled "Clinical value of serum polyunsaturated fatty acids in patients with gastric polyps". They affirm that AA, EPA, DHA, ω-6-C22:5 and the ratio ω6:ω3 differed significantly between control and gastric polyp groups. The authors also investigated the effect of applying certain critical values to the ratio ω6:ω3 to check their relation with the separation between control and polyp groups.

Numerous grammatical and formatting errors are present throughout, affecting its overall quality. The paper needs an in-depth English-language edition. I highly recommend that the authors consider employing the services of a professional editing service to correct them and enhance the overall presentation of the manuscript.

Please be careful when reporting the authors' affiliations, they are numbered from 1 to 4 but only the number 1 and number 3 with the author names.

The abbreviations should be defined the first time used and, after that, they should be used to facilitate the reading. In addition, we recommend using an appropriate term when referring to the analytes measured, you are not measuring only levels of PUFA acids. Please, reconsider the sentence in line 25. The authors use the same term in other places in the manuscript and it is not correct. Moreover, I highly recommend the use of standard terminology and following the same rules for all the fatty acids, for example, it could be C18:2 (n-6), C20:4 (n-6), C18:3 (n=3), etc, but there are other possibilities too. In addition, authors can also add the compound common names. Thus, the reader can distinguish between omega-3, omega-6 and other type of compounds.

The raw data has been shared in supplemental materials, but the group to which the participants belong is not included, nor is the meaning of the colours used to highlight some rows, and some information is missed such as the age of some participants.

The sentence in the abstract and text indicating “moreover, when 6 and 10 were the critical value for the ratio w6:w3, the occurrence of polyps was closely related to the ratio of w6 to w3” needs to be rephrased to make it easy to understand.

The literature cited was reasonable, but it should be improved, for example, by introducing references for the authors' affirmations in lines 48 to 50.

The term “participants” will be more appropriate than “patients” when healthy controls are included in a study. For example, in line 61 “enrolled patients” can be changed to “enrolled participants”. I encourage the authors to check this along with the main text.

Experimental design

A) The collection and storage of the serum need to be described.
B) In the PUFAs detection (please, change this title, not only PUFAs are measured):

B1) I recommend modifying the sentence in lines 72-73 “The 11 kinds of PUFAs levels were detected” It sounds as if you stratify by levels, I think this is not what the authors intend to say.
B2) The full details of the method extraction, instrument parameters and validation data need to be provided. The reference to the manufacturer´s instructions is not enough. Please provide full instrumental details and at least a basic set of validation data, including for example:
a. The lipids extraction method. Full step-by-step procedure for sample extraction and preparation before performing LS-MS/MS.
b. LC-MS/MS. Full Instrument parameters
c. Internal standards used.
d. Calibration
e. Recovery
f. Limits of detection and quantification

C) Statistical analysis

Authors apply the Kruskal-Wallis test for comparison between multiple groups. In this test, the null hypothesis (H0) is that median concentrations across the three groups are equal. Since the p-values of the test (ranging from 0.060 to 0.596) are not less than 0.05, it fails to reject the H0 hypothesis, so authors do not have sufficient evidence to say that there is a statistically significant difference between the median fatty acid concentration across the three groups (the same with the ratio ω6:ω3). Given these results, the subsequent use of the Mann-Whitney test is not justified. Therefore, authors should not affirm that they found significant differences in the AA, EPA, omega-6 C22:5, DHA and ratio ω6:ω3.

Validity of the findings

The research question is original, and there is not much information on the field, but I am sorry to say that the presented data demonstrate that there are no significant differences among the three groups of participants. The results presented in Table 1 do not appear to support the main conclusion proposed by the authors. In addition, the information is contradictory: in lines 87-89 authors said “A total of 323 patients….were enrolled “Data were analyzed for 92 in the gastric polyp group, 116 in the gastritis group and 115 in the control group” whereas in Table 1 information is showed for 92 in the gastric polyp group, 163 in the inflammatory group (I do not know it this is the gastritis group) and 67 in the control group. In Table 2 there are also 67 participants in the control group. In Supplemental Table registries of 368 patients are included.

Additional comments

1) Lines 66-67. The authors classify gastric polyp patients into 3 groups: inflammatory polyps, hyperplastic, and fundic gland polyps. Based upon their histomorphology gastric polyps may be divided in general into hyperplastic polyps, adenomatous polyps, and fundic gland polyps. The term ‘inflammatory polyps’ is a commonly used misnomer for hyperplastic polyps (Waldum H and Fossmark R, Int J Mol Sci. 2021;22(12):6548. doi: 10.3390/ijms22126548; Goddard AF et al. Gut. 2010;59(9):1270-6. DOI: 10.1136/gut.2009.182089). If possible, change it to a more appropriate term.

2) The concentration units are not indicated in the tables.

3) Lines 91-92 indicate that “the median of ω6/ω3 value was highest in the gastritis group, followed by the control group, and the ratio was lowest in the gastric polyp group” whereas in lines 159-160, authors indicate that “the ω3/ω6 value in gastric polyps patients was significantly lower than in healthy people. These two sentences indicate opposite results, they are contradictory. See also lines 163-164.

4) Lines 98-103. Differences among single polyp and multiple polyp subgroups, and concerning histopathologic classification. You should indicate “data not shown” because this information is not included in the Tables provided.

Reviewer 2 ·

Basic reporting

In summary, the authors present an interesting study evaluating the association of concentration of serum Polyunsaturated fatty acids with the endoscopic evidence of gastric polyps. However, some concerns arise regarding the study design and data analysis (why I would focus here rather than on the manuscript's content). I suggest addressing these major issues before acceptance of the manuscript.

Experimental design

1 - The authors must specify and clarify the nature of the study (e.g., is this a cross-sectional study using stored serum samples?), and a study flow diagram should be included.
2 - The authors mention that the patients were not receiving treatment for high cholesterol or high triglycerides. However, they failed to present the groups' baseline clinical and demographic/anthropometric characteristics. How many patients had a BMI >25mg/m2 or were obese? - did they have any significant systemic of GI inflammatory conditions? Any history of familiar disease? A new table must address this concern.
3 - The authors must clarify in the methods section how pathological/histological examination was performed (by one or more pathologists?). Was it blinded? Etc... whyn't all the subjects have pathological reports?
4- Was data analysis blinded?, or serum analyses were performed knowing the subject's "status"? - better information on the study design is needed.
5 - Include in the data analysis how it was determined that parametric assumptions were not satisfied (Q-Q plots of all covariable? Shapiro Wilk tests?
6 - Include in the data analysis section why 6 and 10 were the critical values for evaluating the ratio of Ë-6: Ë-3; also, it would be better to perform an ROC analysis and establish an optimal cutoff with the associated specificity and sensitivity rather than using random cutoff testing differences in proportions?
7 - The methods section should clearly specify the definition of normal subjects, subjects with inflammatory conditions (gastritis), and subjects with gastric polyps.

Validity of the findings

1- As mentioned above, it is easier to drawdraw conclusions by considering all these potential variables as potential confounders (e.g., BMI, personal history, or family history of metabolic disorders). It will be adequate to redo statistical analysis considering this new data.
2 - I am not sure if this affirmation "In our study, some fats, especially the ratio of Ë-6 to Ë-3, played an
important role in the development of gastric polyps and could be used as early screening markers
For gastric polyps," can be used; this expression overstates the real findings (line 32).
3 - Do the authors perform any correlation analysis between serum concentrations of PUFAs and the number of polyps documented in the EGD? or the Polyp Sizes? that could add essential data to their study rather than comparing only single polyps vs. multiple polyps.
4- I suggest adding one paragraph in the discussion of the authors expanding the information on how these findings will impact clinical practice (e.g., treatment or early detection of gastric polyps? different diseases, metabolic profiles?).
5 - Authors should recognize the study limitations at the end of the discussion.
6 - The supplementary material only include gender and age as covariates besides the serum concentrations of PUFAs, the datase do not include the classification according to the presence or not of gastric polips which limitates the accuracy and reproducibility of the statistical analysis. Not clear what the red and yellow lines means.

Additional comments

Minor concerns include:
1 - English should be carefully reviewed. Multiple obvious grammar errors were detected along the document (e.g., lines 29-30, 50-52, 60-61, etc...). For example, line 137 is hardly readable: "Our study showed that compared with healthy people, the median level of ALA in gastric polyps patients were similar, while the upper and lower quartiles are decreased, which suggests a compensatory increase in ALA transformation may appear."
2 -The abstract should highlight the results (desirable to include quantitative data) and the conclusions.
3- I suggest making a transposition in Table 1, "The levels of PUFAs in different groups"; this will facilitate the interpretation of results.
4- Affirmation in line 117 is not supported by any reference.
5 - Lines 105-108 should be part of the introduction rather than part of the discussion.

---

## Round 0.2 · accepted · Accept

All issues pointed by the reviewers were adequately addressed and the manuscript was revised accordingly. Therefore, I am accepting revised version.